# Bayesian Attention Modules

**Xinjie Fan**[*,1]**, Shujian Zhang**[*,1]**, Bo Chen**[2]**, and Mingyuan Zhou**[1]

[1]The University of Texas at Austin and [2]Xidian University

xfan@utexas.edu, szhang19@utexas.edu,
bchen@mail.xidian.edu.cn, mingyuan.zhou@mccombs.utexas.edu

## Abstract

Attention modules, as simple and effective tools, have not only enabled deep neural networks to achieve state-of-the-art results in many domains, but also enhanced their interpretability. Most current models use deterministic attention modules due to their simplicity and ease of optimization. Stochastic counterparts, on the other hand, are less popular despite their potential benefits. The main reason is that stochastic attention often introduces optimization issues or requires significant model changes. In this paper, we propose a scalable stochastic version of attention that is easy to implement and optimize. We construct simplex-constrained attention distributions by normalizing reparameterizable distributions, making the training process differentiable. We learn their parameters in a Bayesian framework where a data-dependent prior is introduced for regularization. We apply the proposed stochastic attention modules to various attention-based models, with applications to graph node classification, visual question answering, image captioning, machine translation, and language understanding. Our experiments show the proposed method brings consistent improvements over the corresponding baselines.

## 1 Introduction

Attention modules, aggregating features with weights obtained by aligning latent states, have become critical components for state-of-the-art neural network models in various applications, such as natural language processing [1–4], computer vision [5, 6], graph analysis [7, 8], and multi-modal learning [9–12]. They have been proven to be effective in not only being combined with other types of neural network components, such as recurrent [1, 9] and convolutional units [5, 13], but also being used to build a stand-alone architecture [2, 6, 14]. Besides boosting the performance, using them also often helps aid model visualization and enhance interpretability [1, 9].

While the attention mechanism provides useful inductive bias, attention weights are often treated as deterministic rather than random variables. Consequently, the only source of randomness lies at the model output layer. For example, in classification models, the randomness is in the final logistic regression layer, while in discrete sequence generation, it is in the conditional categorical output layer. However, a single stochastic output layer is often insufficient in modeling complex dependencies [15]. The idea of augmenting deterministic neural networks with latent random variables has achieved success in many fields to model highly structured data, such as texts [16, 17], speeches [15, 18, 19], natural language sequences [20–23], and images [24–26]. Such modification may not only boost the performance, but also provide better uncertainty estimation [27, 28].

As attention weights can be interpreted as alignment weights, it is intuitive to connect the attention module with latent alignment models [9, 29, 30], where latent alignment variables are stochastic and the objective becomes a lower bound of the log marginal likelihood. Making the attention weights stochastic and learning the alignment distribution in a probabilistic manner brings several potential

---

[*] Equal contribution. Corresponding to: mingyuan.zhou@mccombs.utexas.edu

advantages. First, adding latent random variable enhances the model's ability to capture complicated dependencies in the target data distribution. Second, we are able to adopt Bayesian inference, where we may build our prior knowledge into prior regularization on the attention weights and utilize posterior inference to provide a better basis for model analysis and uncertainty estimation [28, 29].

Most current work on stochastic attention focus on hard attention [9, 29, 31], where the attention weights are discrete random variables sampled from categorical distributions. However, standard backpropagation no longer applies to the training process of such models and one often resorts to a REINFORCE gradient estimator [32], which has large variance. Such models generally underperform their deterministic counterparts, with a few exceptions, where a careful design of baselines and curriculum learning are required [9, 29, 30]. While attending to multiple positions at one time is intuitively more preferable, probabilistic soft attention is less explored. Bahuleyan et al. [33] propose to use the normal distribution to generate the attention weights, which are hence possibly negative and do not sum to one. Deng et al. [29] consider sampling attention weights from the Dirichlet distribution, which is not reparameterizable and hence not amenable to gradient descent based optimization.

In this paper, we propose Bayesian attention modules where the attention weights are treated as latent random variables, whose distribution parameters are obtained by aligning keys and queries. We satisfy the simplex constraint on the attention weights, by normalizing the random variables drawn from either the Lognormal or Weibull distributions. Both distributions generate non-negative random numbers that are reparameterizable. In this way, the whole training process can be made differentiable via the reparameterization trick. We introduce a contextual prior distribution whose parameters are functions of keys to impose a Kullback–Leibler (KL) divergence based regularization. To reduce the variance of gradient estimation, we pick the prior distribution such that the KL term can be rewritten in a semi-analytic form, $i.e.$, an expectation of analytic functions.

Compared to previous stochastic attentions, our method is much simpler to implement, requires only a few modifications to standard deterministic attention, is stable to train, and maintains good scalability, thereby making it attractive for large-scale deep learning applications. We evaluate the proposed stochastic attention module on a broad range of tasks, including graph node classification, visual question answering, image captioning, machine translation, and language understanding, where attention plays an important role. We show that the proposed method consistently outperforms baseline attention modules and provides better uncertainty estimation. Further, we conduct a number of ablation studies to reason the effectiveness of the proposed model.

## 2   Preliminaries on attention modules

In this section, we briefly review the standard deterministic soft attention modules that have been widely used in various neural networks.

**Basic module:** Consider $n$ key-value pairs, packed into a key matrix $K \in \mathbb{R}^{n \times d_k}$ and a value matrix $V \in \mathbb{R}^{n \times d_v}$, and $m$ queries packed into $Q \in \mathbb{R}^{m \times d_k}$, where the dimensions of queries and keys are both equal to $d_k$. Depending on the applications, key, value, and query may have different meanings. For example, in self-attention layers [2], key, value, and query are all from the same source, $i.e.$, the output of the previous layer and in this case $m$ equals to $n$. In encoder-decoder attention layers, the queries come from the decoder layer, while the keys and values come from the output of the encoder [1, 2, 9]. When attention is used for multi-modal cases, the queries often come from one modality while the keys and values come from the other [11].

Attention modules make use of keys and queries to obtain *deterministic* attention weights $W$, which are used to aggregate values $V$ into output features $O = WV \in \mathbb{R}^{m \times d_v}$. Specifically, $W$ is obtained through a softmax function across the key dimension as $W = \mathrm{softmax}(f(Q, K)) \in \mathbb{R}^{m \times n}$, so that it is a non-negative matrix with each row summing to one. Thus if we denote $\Phi = f(Q, K)$, then

$$W_{i,j} = \frac{\exp(\Phi_{i,j})}{\sum_{j'=1}^{n} \exp(\Phi_{i,j'})}. \tag{1}$$

Intuitively, the scale of element $W_{i,j}$ represents the importance of the $j$th key to the $i$th query, and the neural network should learn $Q$ and $K$ such that $W$ gives higher weights to more important features. There are many choices of the alignment score function $f$, including scaled dot-product [2, 3], additive attention [1, 9, 10], and several other variations [13, 34, 35].

**Multi-head and multi-layer attention:** Multi-head attention is proposed to attend to information from different representation subspaces [2], where queries, keys, and values are linearly projected $H$ times by $H$ different projection matrices, producing $H$ output values that are concatenated as the final attention layer output. One may then stack attention layers by placing one on top of another, leading to deep attention modules [2, 11]. For a deep attention module with $L$ attention layers and $H$ heads for each layer, the output of the $l$th layer would be $O^l = [W^{l,1}V^{l,1}, ..., W^{l,H}V^{l,H}]$, where $W^{l,h} = \text{softmax}(f(Q^{l,h}, K^{l,h}))$, $Q^{l,h} = Q^l M_Q^{l,h}$, $K^{l,h} = K^l M_K^{l,h}$, and $V^{l,h} = V^l M_V^{l,h}$ for $h = 1, ..., H$, and the $M$'s are parametric matrices that the neural network needs to learn. Then the output of the $l$th attention layer, $O^l$, is fed into the next attention layer (possibly after some transformations) and the queries, keys, and values of the $(l+1)$th layer would be functions of $O^l$.

## 3 Bayesian attention modules: a general recipe for stochastic attention

We suggest a general recipe for stochastic attention: 1) treat attention weights as data-dependent local random variables and learn their distributions in a Bayesian framework, 2) use normalized reparametrizable distributions to construct attention distributions over simplex, and 3) use a key-based contextual prior as regularization.

### 3.1 Learning attention distributions in a Bayesian way

Consider a supervised learning problem with training data $\mathcal{D} := \{\boldsymbol{x}_i, \boldsymbol{y}_i\}_{i=1}^N$, where we model the conditional probability $p_{\boldsymbol{\theta}}(\boldsymbol{y}_i \,|\, \boldsymbol{x}_i)$ using a neural network parameterized by $\boldsymbol{\theta}$, which includes the attention projections $M$'s. For notational convenience, below we drop the data index $i$. Using vanilla attention modules, the mapping from $\boldsymbol{x}$ to the likelihood $p_{\boldsymbol{\theta}}(\boldsymbol{y}|\boldsymbol{x})$ is deterministic, so the whole model is differentiable meaning that it is tractable to directly maximize the likelihood.

Now, we turn the mapping from queries and keys to attention weights $W$ stochastic. Instead of using deterministic weights obtained from queries and keys to aggregate values, we treat $W = \{W^{l,h}\}_{l=1:L,h=1:H}$ as a set of data-dependent local latent variables sampled from $q_{\boldsymbol{\phi}}$, which can be parameterized by some functions of queries and keys. Intuitively, we argue that this distribution can be viewed as a variational distribution approximating the posterior of local attention weights $W$, under a Bayesian model, given the data $\boldsymbol{x}, \boldsymbol{y}$. Therefore, we can learn $q_{\boldsymbol{\phi}}$ with amortized variational inference [24]. Note that, unlike Deng et al. [29] and Lawson et al. [30], we do not enforce $q_{\boldsymbol{\phi}}$ to be dependent on $\boldsymbol{y}$, which might not be available during testing. Instead, we use the queries and keys in standard attention modules to construct $q_{\boldsymbol{\phi}}$, so $q_{\boldsymbol{\phi}}$ depends on $\boldsymbol{x}$ only or both $\boldsymbol{x}$ and the part of $\boldsymbol{y}$ that has already been observed or generated by the model. For example, in visual question answering or graph node classification, $q_{\boldsymbol{\phi}}$ only depends on input $\boldsymbol{x}$. While in sequence generation, like image captioning or machine translation, $q_{\boldsymbol{\phi}}$ could be dependent on the observed part of $\boldsymbol{y}$ as the queries come from $\boldsymbol{y}$.

Constructing variational distribution in such a way has several advantages. First, as we will show in the next section, by utilizing keys and values, transforming a set of deterministic attention weights into an attention distribution becomes straightforward and requires minimal changes to standard attention models. We can even easily adapt pretrained standard attention models for variational finetuning (shown in Section 4.5). Otherwise, building an efficient variational distribution often requires domain knowledge [29] and case by case consideration. Second, due to a similar structure as standard attention modules, $q_{\boldsymbol{\phi}}$ introduces little additional memory and computational cost, for which we provide a complexity analysis in Section 3.4. Third, as keys and values are available for both training and testing, we can use the variational distribution $q_{\boldsymbol{\phi}}$ during testing. By contrast, previous works [29, 30] enforce $q_{\boldsymbol{\phi}}$ to include information not available during testing, restricting its usage at the testing time. Further, as keys and queries depend on the realization of attention weights in previous layers, this structure naturally allows *cross-layer dependency* between attention weights in different layers so that it is capable of modeling complex distributions.

Consider a Bayesian model, where we have prior $p_{\boldsymbol{\eta}}(W)$ and likelihood $p_{\boldsymbol{\theta}}(\boldsymbol{y} \,|\, \boldsymbol{x}, W)$ that share a common structure with vanilla deterministic soft attention. We learn the distribution $q_{\boldsymbol{\phi}}$ by minimizing $\text{KL}(q_{\boldsymbol{\phi}}(W)||p(W \,|\, \boldsymbol{x}, \boldsymbol{y}))$, the KL divergence from the posterior distribution of $W$ given $\boldsymbol{x}$ and $\boldsymbol{y}$ to $q_{\boldsymbol{\phi}}$. With amortized variational inference, it is equivalent to maximizing $\mathcal{L}_{\mathcal{D}} = \sum_{(\boldsymbol{x},\boldsymbol{y}) \in \mathcal{D}} \mathcal{L}(\boldsymbol{x}, \boldsymbol{y})$, an evidence lower bound (ELBO) [29, 36, 37] of the intractable log marginal

likelihood $\sum_{(\boldsymbol{x},\boldsymbol{y})\in\mathcal{D}} \log p(\boldsymbol{y} \mid \boldsymbol{x}) = \sum_{(\boldsymbol{x},\boldsymbol{y})\in\mathcal{D}} \log \int p_{\boldsymbol{\theta}}(\boldsymbol{y} \mid \boldsymbol{x}, W) p_{\boldsymbol{\eta}}(W) dW$, where

$$\mathcal{L}(\boldsymbol{x},\boldsymbol{y}) := \mathbb{E}_{q_{\boldsymbol{\phi}}(W)}\left[\log p_{\boldsymbol{\theta}}(\boldsymbol{y} \mid \boldsymbol{x}, W)\right] - \mathrm{KL}(q_{\boldsymbol{\phi}}(W) \| p_{\boldsymbol{\eta}}(W)) = \mathbb{E}_{q_{\boldsymbol{\phi}}(W)}\left[\log \frac{p_{\boldsymbol{\theta}}(\boldsymbol{y} \mid \boldsymbol{x}, W) p_{\boldsymbol{\eta}}(W)}{q_{\boldsymbol{\phi}}(W)}\right].$$

Learning attention distribution $q_{\boldsymbol{\phi}}$ via amortized variational inference provides a natural regularization for $q_{\boldsymbol{\phi}}$ from prior $p_{\boldsymbol{\eta}}$, where we can inject our prior beliefs on attention distributions. We will show we can parameterize the prior distribution with keys, so that the prior distribution can be data-dependent and encode the importance information of each keys. Meanwhile, we can update $\boldsymbol{\theta}$ and $\boldsymbol{\eta}$ to maximize the ELBO. As $q_{\boldsymbol{\phi}}$ becomes closer to the posterior, the ELBO becomes a tighter lower bound.

### 3.2 Reparameterizable attention distributions

A challenge of using stochastic attention weights is to optimize their distribution parameters. Existing methods [9, 29, 31] construct attention distributions in a way that standard backpropagation based training no longer applies. Without carefully customizing a training procedure for each specific task, it is generally hard to learn such distributions. Below we introduce reparameterizable soft stochastic attentions that allow effectively optimizing the distribution parameters in a simple and general way.

Our goal is to construct a reparameterizable attention distribution $q_{\boldsymbol{\phi}}$ over the simplex, *i.e.*, $W_{i,j}^{l,h} \geq 0$ and $\sum_j W_{i,j}^{l,h} = 1$. While the Dirichlet distribution, satisfying the simplex constraint and encouraging sparsity, appears to be a natural choice, it is not reparameterizable and hence not amenable to gradient descent based optimization. Here, we consider satisfying the simplex constraint by normalizing random variables drawn from non-negative reparameterizable distributions. In particular, we consider the Weibull and Lognormal distributions. We choose them mainly because they both lead to optimization objectives that are simple to optimize, as described below.

**Weibull distribution:** The Weibull distribution $S \sim \text{Weibull}(k, \lambda)$ has probability density function (PDF) $p(S \mid k, \lambda) = \frac{k}{\lambda^k} S^{k-1} e^{-(S/\lambda)^k}$, where $S \in \mathbb{R}_+$. Its expectation is $\lambda\Gamma(1 + 1/k)$ and variance is $\lambda^2\left[\Gamma\left(1 + 2/k\right) - \left(\Gamma\left(1 + 1/k\right)\right)^2\right]$. It is reparameterizable as drawing $S \sim \text{Weibull}(k, \lambda)$ is equivalent to letting $S = \tilde{g}(\epsilon) := \lambda(-\log(1 - \epsilon))^{1/k}$, $\epsilon \sim \text{Uniform}(0, 1)$. It resembles the gamma distribution, and with $\gamma$ denoted as the Euler–Mascheroni constant, the KL divergence from the gamma to Weibull distributions has an analytic expression [17] as

$$\mathrm{KL}\big(\text{Weibull}(k, \lambda)\|\text{Gamma}(\alpha, \beta)\big) = \frac{\gamma\alpha}{k} - \alpha\log\lambda + \log k + \beta\lambda\Gamma(1 + \tfrac{1}{k}) - \gamma - 1 - \alpha\log\beta + \log\Gamma(\alpha).$$

**Lognormal distribution:** The Lognormal distribution $S \sim \text{Lognormal}(\mu, \sigma^2)$ has PDF $p(S \mid \mu, \sigma) = \frac{1}{S\sigma\sqrt{2\pi}}\exp\left[-\frac{(\log S - \mu)^2}{2\sigma^2}\right]$, where $S \in \mathbb{R}_+$. Its expectation is $\exp(\mu + \sigma^2/2)$ and variance is $[\exp(\sigma^2) - 1]\exp(2\mu + \sigma^2)$. It is also reparameterizable as drawing $S \sim \text{Lognormal}(\mu, \sigma^2)$ is equivalent to letting $S = \tilde{g}(\epsilon) = \exp(\epsilon\sigma + \mu)$, $\epsilon \sim \mathcal{N}(0, 1)$. The KL divergence is analytic as

$$\mathrm{KL}(\text{Lognormal}(\mu_1, \sigma_1^2)\|\text{Lognormal}(\mu_2, \sigma_2^2)) = \log\frac{\sigma_2}{\sigma_1} + \frac{\sigma_1^2 + (\mu_1 - \mu_2)^2}{2\sigma_2^2} - 0.5.$$

Sampling $S_{i,j}^{l,h}$ from either the Weibull or Lognormal distribution, we obtain the simplex-constrained random attention weights $W$ by applying a normalization function $\bar{g}$ over $S$ as $W_i^{l,h} = \bar{g}(S_i^{l,h}) := S_i^{l,h} / \sum_j S_{i,j}^{l,h}$. Note $W$ is reparameterizable but often does not have an analytic PDF.

**Parameterizing variational attention distributions:** To change the deterministic mapping from the queries and keys to attention weights to a stochastic one, we use queries and keys to obtain the distribution parameters of unnormalized attention weights $S$, which further define the variational distribution of the normalized attention weights $W$.

With the *Weibull* distribution, we treat $k$ as a global hyperparameter and let $\lambda_{i,j}^{l,h} = \exp(\Phi_{i,j}^{l,h})/\Gamma(1 + 1/k)$, and like before $\Phi^{l,h} = f(Q^{l,h}, K^{l,h})$. Then, we sample $S_{i,j}^{l,h} \sim \text{Weibull}(k, \lambda_{i,j}^{l,h})$, which is the same as letting $S_{i,j}^{l,h} = \exp(\Phi_{i,j}^{l,h})\frac{(-\log(1 - \epsilon_{i,j}^{h,l}))^{1/k}}{\Gamma(1 + 1/k)}$, $\epsilon_{i,j}^{h,l} \sim \text{Uniform}(0, 1)$. With the *Lognormal* distribution, we treat $\sigma$ as a global hyperparameter and let $\mu_{i,j}^{l,h} = \Phi_{i,j}^{l,h} - \sigma^2/2$. Then, we sample $S_{i,j}^{l,h} \sim \text{Lognormal}(\mu_{i,j}^{l,h}, \sigma^2)$, which is the same as letting $S_{i,j}^{l,h} = \exp(\Phi_{i,j}^{l,h})\exp(\epsilon_{i,j}^{h,l}\sigma - \sigma^2/2)$, $\epsilon_{i,j}^{h,l} \sim \mathcal{N}(0, 1)$. Note our parameterizations ensure that $\mathbb{E}[S_{i,j}^{l,h}] = \exp(\Phi_{i,j}^{l,h})$. Therefore,

if, instead of sampling $S_{i,j}^{l,h}$ from either distribution, we use its expectation as a substitute, then the mapping becomes equivalent to that of vanilla soft attention, whose weights are defined as in (1). In other words, if we let $k$ of the Weibull distribution go to infinity, or $\sigma$ of the Lognormal distribution go to zero, which means the variance of $S_{i,j}^{l,h}$ goes to zero and the distribution becomes a point mass concentrated at the expectation, then the proposed stochastic soft attention reduces to deterministic soft attention. Therefore, the proposed stochastic soft attention can be viewed as a generalization of vanilla deterministic soft attention.

We have now constructed $q_\phi$ to be a reparameterizable distribution $W = g_\phi(\epsilon) := \bar{g}(\tilde{g}_\phi(\epsilon))$, where $\epsilon$ is a collection of $i.i.d.$ random noises with the same size as $W$. To estimate the gradient of the ELBO, however, we need either the analytic forms of both $p_\eta$ and $q_\phi$, or the analytic form of the KL term, neither of which are available. In the next section, we show how to work around this issue by imposing the KL regularization for latent variable $S$ before normalization and decomposing the joint distribution into a sequence of conditionals.

### 3.3 Contextual prior: key-dependent Bayesian regularization

In the ELBO objective, there is a built-in regularization term, $i.e.$, the KL divergence from the prior distribution $p_\eta$ to variational distribution $q_\phi$. To estimate the gradients, we need to evaluate $q_\phi(W)$ and $p_\eta(W)$ for given attention weights $W$. We note that even though the analytic form of $q_\phi(W)$ is not available, $q_\phi(S) = \prod_{l=1}^{L} q_\phi(S_l \,|\, S_{1:l-1})$ is a product of analytic PDFs (Weibull or Lognormal) for unnormalized weights $S$, so we rewrite the ELBO in terms of $S$ (we keep using $q_\phi, p_\eta$ for $S$ as the distribution of $W$ is defined by $S$),

$$\mathcal{L}(\boldsymbol{x}, \boldsymbol{y}) := \mathbb{E}_{q_\phi(S)} \left[ \log p_{\boldsymbol{\theta}}(\boldsymbol{y} \,|\, \boldsymbol{x}, S) \right] - \mathrm{KL}(q_\phi(S) || p_\eta(S)) \tag{2}$$

Note the KL divergence, as shown in Section 3.2, can be made analytic and hence it is natural to use either the gamma or Lognormal distribution to construct $p_\eta(S)$. Regardless of whether the gamma or Lognormal is used to construct $p_\eta(S)$, due to the dependencies between different stochastic attention layers, we do not have analytic expressions for $\mathrm{KL}(q_\phi(S) || p_\eta(S))$. Fortunately, as shown in Lemma 1, by decomposing the joint into a sequence of conditionals and exploiting these analytic KL divergence expressions, we can express each KL term in a semi-analytic form. According to the Rao-Blackwellization theorem [38], we can reduce the Monte Carlo estimation variance by plugging in the analytic part.

**Lemma 1.** *The KL divergence from the prior to variational distributions is semi-analytic as*

$$\mathrm{KL}(q_\phi(S) || p_\eta(S)) = \sum\nolimits_{l=1}^{L} \mathbb{E}_{q_\phi(S_{1:l-1})} \underbrace{\mathrm{KL}(q_\phi(S_l | S_{1:l-1}) || p_\eta(S_l | S_{1:l-1}))}_{analytic} \tag{3}$$

*Proof.*

$$\begin{aligned}
\mathrm{KL}(q_\phi(S) || p_\eta(S)) &= \mathbb{E}_{q_\phi(S)} \left[ \sum_{l=1}^{L} (\log q_\phi(S_l | S_{1:l-1}) - \log p_\eta(S_l | S_{1:l-1})) \right] \\
&= \sum_{l=1}^{L} \mathbb{E}_{q_\phi(S)} \left[ \log q_\phi(S_l | S_{1:l-1}) - \log p_\eta(S_l | S_{1:l-1}) \right] \\
&= \sum_{l=1}^{L} \mathbb{E}_{q_\phi(S_{1:l-1})} \mathbb{E}_{q_\phi(S_l | S_{1:l-1})} \left[ \log q_\phi(S_l | S_{1:l-1}) - \log p_\eta(S_l | S_{1:l-1}) \right].
\end{aligned} \tag{4}$$

$\square$

**Key-based contextual prior:** Instead of treating the prior as a fixed distribution independent of the input $\boldsymbol{x}$, here we make the prior depend on the input through keys. The motivation comes from our application in image captioning. Intuitively, given an image (keys), there should be a *global* prior attention distribution over the image, indicating the importance of each part of the image even before the caption generation process. Based on the prior distribution, the attention distribution can be updated *locally* using the current state of generation (queries) at the each step (see Figure 1). This intuition can be extended to the general attention framework, where the prior distribution encodes the *global* importance of each keys shared by all queries, while the posterior encodes the *local* importance

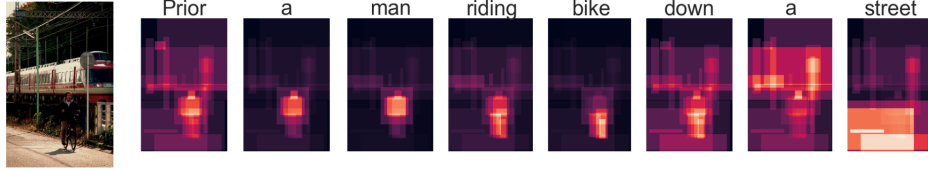

| | Prior | a | man | riding | bike | down | a | street |

Figure 1: Visualization of attention weight samples from contextual prior distribution and variational distributions at each step for image captioning. Given the image, prior attention distribution over the image areas encodes the importance of each part before the caption generation process. Based on the prior distribution, the attention distribution can be updated at the each step using the current state of generation.

of each keys for each query. To obtain the prior parameters, we take a nonlinear transformation of the key features, followed by a softmax to obtain positive values and enable the interactions between keys. Formally, let $\Psi^{l,h} = \text{softmax}(F_2(F_{NL}(F_1(K^{l,h})))) \in \mathbb{R}^{n \times 1}$, where $F_1$ is linear mapping from $\mathbb{R}^{d_k}$ to a hidden dimension $\mathbb{R}^{d_{\text{mid}}}$, $F_2$ is linear mapping from $\mathbb{R}^{d_{\text{mid}}}$ to $\mathbb{R}$, and $F_{NL}$ denotes a nonlinear activation function, such as ReLU [39]. With the gamma prior, we treat $\beta$ as a hyperparameter and let $\alpha_{i,j}^{l,h} = \Psi_{i,1}^{l,h}$. With the Lognormal, we treat $\sigma$ as a hyperparameter and let $\mu_{i,j}^{l,h} = \Psi_{i,1}^{l,h}$. Following previous work [40], we add a weight $\lambda$ to the KL term and anneal it from a small value to one.

### 3.4 Putting it all together

Combining (2) and (3) and using reparameterization, we have $\mathcal{L}_\lambda(\boldsymbol{x}, \boldsymbol{y}) = \mathbb{E}_{\boldsymbol{\epsilon}}[\mathcal{L}_\lambda(\boldsymbol{x}, \boldsymbol{y}, \boldsymbol{\epsilon})]$, where

$$\mathcal{L}_\lambda(\boldsymbol{x}, \boldsymbol{y}, \boldsymbol{\epsilon}) = \log p_{\boldsymbol{\theta}}(\boldsymbol{y} \mid \boldsymbol{x}, \tilde{g}_{\boldsymbol{\phi}}(\boldsymbol{\epsilon})) - \lambda \sum_{l=1}^{L} \underbrace{\text{KL}(q_{\boldsymbol{\phi}}(S_l \mid \tilde{g}_{\boldsymbol{\phi}}(\boldsymbol{\epsilon}_{1:l-1})) || p_{\boldsymbol{\eta}}(S_l \mid \tilde{g}_{\boldsymbol{\phi}}(\boldsymbol{\epsilon}_{1:l-1})))}_{\textit{analytic}}. \quad (5)$$

To estimate the gradient of $\mathcal{L}_\lambda(\boldsymbol{x}, \boldsymbol{y})$ with respect to $\boldsymbol{\phi}, \boldsymbol{\theta}, \boldsymbol{\eta}$, we compute the gradient of $\mathcal{L}_\lambda(\boldsymbol{x}, \boldsymbol{y}, \boldsymbol{\epsilon})$, which is a Monte Carlo estimator with one sample of $\boldsymbol{\epsilon}$. This way provides unbiased and low-variance gradient estimates (see the pseudo code in Algorithm 1 in Appendix).

At the testing stage, to obtain point estimates, we adopt the common practice of approximating the posterior means of prediction probabilities by substituting the latent variables by their posterior expectations [41]. To calibrate estimation uncertainties, we draw multiple posterior samples, each of which produces one posterior prediction probability sample.

**Complexity analysis:** Our framework is computationally and memory efficient due to parameter sharing between the variational, prior, and likelihood networks. Extra memory cost comes from the contextual prior network which, for a single layer and single head attention, is of scale $O(d_k d_{\text{mid}})$. This is insignificant compared to the memory scale of $M$'s, $O(d_k d_v + d_v d_v)$, as $d_{\text{mid}}$ is as small as $10 \ll d_v$. Meanwhile, the additional computations involve the sampling process and computing the KL term which is of scale $O(mn)$. Computing the contextual prior is of scale $O(n d_k d_{\text{mid}})$. All above is inconsiderable compared to the computational scale of deterministic attentions, $O(mn d_k d_v)$.

## 4 Experiments

Our method can be straightforwardly implemented in any attention based models. To test the general applicability of our method, we conduct experiments on a wide range of tasks where attention is essential, covering graphs (node classification), multi-modal domains (visual question answering, image captioning), and natural language processing (machine translation, language understanding). A variety of attention types appear in these domains, including self, encoder-decoder, and guided attentions. In this section, we summarize the main experimental settings and results, and include the details in Appendix B. All experiments are conducted on a single Nvidia Tesla V100 GPU with 16 GB memory. Python code is available at `https://github.com/zhougroup/BAM`

### 4.1 Attention in graph neural networks

We first adapt our method to graph attention networks (GAT) [7], which leverages deterministic *self-attention* layers to process node-features for graph node classification. The graph structure is encoded in the attention masks in a way that nodes can only attend to their neighborhoods' features in the graph. GAT is computationally efficient, capable of processing graphs of different sizes, and achieves state-of-the-art results on benchmark graphs. We use the same model and experimental

setup as in GAT [7], as summarized in Appendix B.1. We experiment with three benchmark graphs, including Cora, Citeseer, and Pubmed, for node classification in a transductive setting, meaning that training and testing are performed on different nodes of the same graph [42]. We include a summary of these datasets in Table 6 in Appendix. For large and sparse graph datasets like Pubmed, following GAT [7], we implement a sparse version of the proposed method, where sparse tensor operations are leveraged to limit the memory complexity to be linear in the number of edges.

Table 1: Classification accuracy for graphs.

| Attention | Cora | Citeseer | PubMed |
|---|---|---|---|
| GAT | 83.00 | 72.50 | 77.26 |
| BAM (NO KL) | 83.39 | 72.91 | 78.50 |
| BAM-LF | 83.24 | 72.86 | 78.30 |
| BAM-LC | 83.34 | 73.04 | 78.76 |
| BAM-WF | 83.48±0.2 | 73.18±0.3 | 78.50±0.3 |
| BAM-WC | **83.81**±0.3 | **73.52**±0.4 | **78.82**±0.3 |

**Results.** The results are summarized in Table 1. We report the results of soft attention (GAT), and 5 versions of Bayesian Attention Modules (BAM): no KL regularization (NO KL), Lognormal with fixed prior (LF), Lognormal with contextual prior (LC), Weibull and fixed prior (WF), and Weibull and contextual prior (WC). We report the mean classification accuracies on test nodes over 5 random runs, and the standard deviations of BAM-WC. Note the results of GAT are reproduced by running the code provided by the authors (`https://github.com/PetarV-/GAT`). Our results demonstrate that adapting the deterministic soft attention module in GAT to Bayesian attention consistently improves the performance. Weibull distribution performs better that Lognormal, and contextual prior outperforms fixed prior and no prior.

## 4.2 Attention in visual question answering models

We consider a multi-modal learning task, visual question answering (VQA), where a model predicts an answer to a question relevant to the content of a given image. The recently proposed MCAN [11] uses *self-attention* to learn the fine-grained semantic meaning of both the image and question, and *guided-attention* to learn the reasoning between these two modalities. We apply BAM to MCAN and conduct experiments on the VQA-v2 dataset [43], consisting of human-annotated question-answer pairs for images from the MS-COCO dataset [44] (see detailed experiment settings in Appendix B.2). To investigate the model's robustness to noise, we also perturb the input by adding Gaussian noise to the image features [45]. For evaluation, we consider both accuracy and uncertainty, which is necessary here as some questions are so challenging that even human annotators might have different answers. We use a hypothesis testing based Patch Accuracy vs Patch Uncertainty (PAvPU) [46] to evaluate the quality of uncertainty estimation, which reflects whether the model is uncertain about its mistakes. We defer the details of this metric to Appendix B.2.1.

Table 2: Results of different attentions on VQA.

| METRIC | ACCURACY | | PAvPU | |
|---|---|---|---|---|
| DATA | ORIGINAL | NOISY | ORIGINAL | NOISY |
| SOFT | 66.95 | 61.25 | 70.04 | 65.34 |
| BAM-LF | 66.89 | 61.43 | 69.92 | 65.48 |
| BAM-LC | 66.93 | 61.58 | 70.14 | 65.60 |
| BAM-WF | 66.93 | 61.60 | 70.09 | 65.62 |
| BAM-WC | **67.02** ±0.04 | **62.89** ±0.06 | **71.21** ±0.06 | **66.75** ±0.08 |

**Results.** In Table 2, we report the accuracy and uncertainty for both original and noisy data (see complete results in Table 7). In terms of accuracy, BAM performs similarly as soft attention on the original dataset, but clearly outperforms it on the more challenging noisy dataset, showing that stochastic soft attention is more robust to noise than deterministic ones. For uncertainty, as soft attention is deterministic we use the dropout in the model to obtain uncertainty, while for BAM we use both dropout and stochastic attention to obtain uncertainty. We observe that on both original and noisy datasets, BAM has better uncertainty estimations, meaning in general it is more uncertain on its mistakes and more certain on its correct predictions. We provide qualitative analysis for uncertainty estimation by visualizing the predictions and uncertainties of three VQA examples in Figure 2 in

Appendix. We note that we again observe the improvement from using contextual prior and that Weibull also performs better than Lognormal.

## 4.3 Attention in image captioning models

We further experiment a multi-modal sequence generation task, image captioning, where probabilistic attention (hard attention) was found to outperform deterministic ones [9]. Image captioning models map an image $x$ to a sentence $y = (y_1, \ldots, y_T)$ that summarizes the image information. *Encoder-decoder attention* is commonly adopted in state-of-the-art models. During encoding, bottom-up bounding box features [47] are extracted from images by a pretrained Faster R-CNN [48]. At each step of decoding, a weighted sum of bounding box features is injected into the hidden states of an LSTM-based RNN [49, 50] to generate words. The weights are computed by aligning the bounding box features (keys) and hidden states from the last step (queries). We conduct our experiments on MS-COCO [44], following the setup of Luo et al. [51]. For the model architecture, we employ an attention-based model (Att2in) of Rennie et al. [10], which we implement based on the code by Luo et al. [51] and replace the ResNet-encoded features by bounding box features (see details in Appendix B.3). In all experiments, we use maximum likelihood estimation (MLE) for training; we do not consider reinforcement learning based fine-tuning [10, 52, 53], which is beyond the scope of this paper and we leave it as future work. We report four widely used evaluation metrics, including BLEU [54], CIDEr [55], ROUGE [56], and METEOR [57].

Table 3: Comparing different attention modules on image captioning.

| ATTENTION | BLEU-4 | BLEU-3 | BLEU-2 | BLEU-1 | CIDEr | ROUGE | METEOR |
|---|---|---|---|---|---|---|---|
| SOFT[9] | 24.3 | 34.4 | 49.2 | 70.7 | - | - | 23.9 |
| HARD[9] | 25.0 | 35.7 | 50.4 | 71.8 | - | - | 23.0 |
| SOFT (OURS) | 32.2 | 43.6 | 58.3 | 74.9 | 104.0 | 54.7 | 26.1 |
| HARD (OURS) | 26.5 | 37.2 | 51.9 | 69.8 | 84.4 | 50.7 | 23.3 |
| BAM-LC | 32.7 | 44.0 | 58.7 | 75.1 | **105.0** | 54.8 | **26.3** |
| BAM-WC | **32.8**±0.1 | **44.1**±0.1 | **58.8**±0.1 | **75.3**±0.1 | 104.5±0.1 | **54.9**±0.1 | 26.2±0.1 |

**Results.** We incorporate the results of both deterministic soft attention and probabilistic hard attention from Xu et al. [9]. We also report those results based on an improved network architecture used by BAM. Results in Table 3 show that the proposed probabilistic soft attention module (BAM) consistently outperforms the deterministic ones. In our implementation, we observe that it is difficult to make hard attention work well due to the high variance of gradients. Moreover, we experiment on modeling the attention weights as Gaussian distribution directly as in Bahuleyan et al. [33]. Our experiment shows that naively modeling attention weights with Gaussian distribution would easily lead to NAN results, as it allows the attention weights to be negative and not sum to 1. Therefore, it is desirable to model attention weights with simplex constrained distributions. We also experiment with BAM where the KL term is completely sampled and observe that the training becomes very unstable and often lead to NAN results. Therefore, constructing prior and posterior in a way that the KL term is semi-analytic does bring clear advantages to our model. In Figure 1, we visualize the prior attention weights and posterior attention weights at each step of generation, where we can visually see how the posterior attention adapts from prior across steps. Further, we again observe that BAM-WC ourperforms BAM-LC and hence for the following tasks, we focus on using BAM-WC.

## 4.4 Attention in neural machine translation

We also experiment with neural machine translation, where we compare with the variational attention method proposed by Deng et al. [29]. We follow them to set the base model structure and experimental setting (see in Appendix B.4), and adapt their deterministic attention to BAM-WC. We compare the BLEU score [54] of BAM-WC and several variants of variational attention in Deng et al. [29].

**Results.** As shown in Table 4, BAM-WC outperforms deterministic soft attention significantly in BLEU score with same-level computational cost. Moreover, compared to all variants of variational attention [29], BAM-WC achieves better performance with much less training cost, as BAM does not require the training of a completely separate variational network. In Table 8 in Appendix, we compare the run time and number of parameters of BAM and variational attention [26], where we show that BAM achieves better results while being more efficient in both time and memory. It is

Table 4: Results on IWSLT.

| Model | BLEU |
|---|---|
| Soft Attention | 32.77 |
| Variational Relaxed Attention | 30.05 |
| Variational Attention + Enum | 33.68 |
| Variational Attention + Sample | 33.30 |
| BAM-WC (Ours) | **33.81**±0.02 |

interesting to note that in Deng et al. [29] variational relaxed attention (probabilistic soft attention) underperforms variational hard attention, while BAM-WC, which is also probabilistic soft attention, can achieve better results. One of the main reason is that BAM-WC is reparameterizable and has stable gradients, while Deng et al. [29] use the Dirichlet distribution which is not reparameterizable so the gradient estimations still have high variances despite the use of a rejection based sampling and implicit differentiation [58]. Also, we note that, compared to Deng et al. [29], our method is much more general because we do not need to construct the variational distribution on a case-by-case basis.

## 4.5 Attention in pretrained language models

Finally, we adapt the proposed method to finetune deterministic *self-attention* [2] based language models pretrained on large corpora. Our variational distribution parameters use the pretrained parameters from the deterministic models, and we randomly initialize the parameters for contextual prior. Then we finetune BAM for downstream tasks. We conduct experiments on 8 benckmark datasets from General Language Understanding Evaluation (GLUE) [59] and two versions of Stanford Question Answering Datasets (SQuAD) [60, 61]. We leverage the state-of-the-art pretrained model, ALBERT [4], which is a memory-efficient version of BERT [3] with parameter sharing and embedding factorization. Our implementation is based on Huggingface PyTorch Transformer [62] and we use the base version of ALBERT following the same setting [4] (summarized in Appendix B.5).

Table 5: Performance of BAM on GLUE and SQuAD benchmarks.

| | MRPC | CoLA | RTE | MNLI | QNLI | QQP | SST | STS | SQuAD 1.1 | SQuAD 2.0 |
|---|---|---|---|---|---|---|---|---|---|---|
| ALBERT-base | 86.5 | 54.5 | 75.8 | 85.1 | 90.9 | **90.8** | 92.4 | 90.3 | 80.86/88.70 | 78.80/82.07 |
| ALBERT-base+BAM-WC | **88.5** | **55.8** | **76.2** | **85.6** | **91.5** | 90.7 | **92.7** | **91.1** | **81.40/88.82** | **78.97/82.23** |

**Results.** In Table 5, we compare the results of ALBERT, which uses deterministic soft attention finetuned on each dataset with those finetuned with BAM-WC, resuming from the same checkpoints. We observe consistent improvements from using BAM-WC in both GLUE and SQuAD datasets even by only using BAM at the finetuning stage. We leave as future work using BAM at the pretrain stage.

## 5 Conclusion

We have proposed a simple and scalable Bayesian attention module (BAM) that achieves strong performance on a broad range of tasks but requires surprisingly few modifications to standard deterministic attention. The attention weights are obtained by normalizing reparameterizable distributions parameterized by functions of keys and queries. We learn the distributions in a Bayesian framework, introducing a key-dependent contextual prior such that the KL term used for regularization is semi-analytic. Our experiments on a variety of tasks, including graph node classification, visual question answering, image captioning, and machine translation, show that BAM consistently outperforms corresponding baselines and provides better uncertainty estimation at the expense of only slightly increased computational and memory cost. Further, on language understanding benchmarks, we show it is possible to finetune a pretrained deterministic attention with BAM and achieve better performance than finetuning with the original deterministic soft attention. With extensive experiments and ablation studies, we demonstrate the effectiveness of each component of the proposed architecture, and show that BAM can serve as an efficient alternative to deterministic attention in the versatile tool box of attention modules.

## Broader Impact

Attention modules have become critical components for state-of-the-art neural network models in various applications, including computer vision, natural language processing, graph analysis, and multi-modal tasks, to name a few. While we show improvements brought by our work on five representative tasks from a broad range of domains, our framework is general enough that it could be used to improve potentially any attention based models. Also, our framework solves two main issues of previously proposed probabilistic attentions that restrict their popularity, $i.e.$, optimization difficulty and complicated model design. We hope that our work will encourage the community to pay more attention to stochastic attention and study from a probabilistic perspective.

Considering that attention models have been adopted in many machine learning systems, our work could have an important impact on those systems, such as self-driving [63], healthcare [64], and recommender systems [65]. However, there are potential risks of applying such systems in real-life scenario, because the data we encounter in real-life is biased and long-tailed, and also the discrepancy between training data and testing data might be large. Therefore, an undue trust in deep learning models, incautious usage or imprecise interpretation of model output by inexperienced practitioners might lead to unexpected false reaction in real-life and unexpected consequences. However, we see opportunities that our work can help mitigate the risks with uncertainty estimation. Knowing when mistakes happen would enable us to know when to ask for human-aid if needed for real-life applications [66].

## Acknowledgements

X. Fan, S. Zhang, and M. Zhou acknowledge the support of Grants IIS-1812699 and ECCS-1952193 from the U.S. National Science Foundation, the support of NVIDIA Corporation with the donation of the Titan Xp GPU used for this research, and the Texas Advanced Computing Center (TACC) at The University of Texas at Austin for providing HPC resources that have contributed to the research results reported within this paper (URL: `http://www.tacc.utexas.edu`). B. Chen acknowledges the support of the Program for Young Thousand Talent by Chinese Central Government, the 111 Project (No. B18039), NSFC (61771361), Shanxi Innovation Team Project, and the Innovation Fund of Xidian University.

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
