[Supplementary Material]

# Bayesian Attention Modules: Appendix

## A   Algorithm

---
**Algorithm 1:** Bayesian Attention Modules

---
$\boldsymbol{\theta}, \boldsymbol{\eta}, \boldsymbol{\phi} \leftarrow$ Initialze parameters, $t \leftarrow 0$, $\rho \leftarrow$ anneal rate
**repeat**
    $\{\boldsymbol{x}_i, \boldsymbol{y}_i\}_{i=1}^M \leftarrow$ Random minibatch of M datapoints (drawn from full dataset)
    $\{\boldsymbol{\epsilon}_i\}_{i=1}^M \leftarrow$ Random samples
    $\lambda = \mathrm{sigmoid}(t * \rho)$
    Compute gradients $\frac{1}{M} \nabla_{\boldsymbol{\theta}, \boldsymbol{\eta}, \boldsymbol{\phi}} \sum_i \mathcal{L}_\lambda(\boldsymbol{x}_i, \boldsymbol{y}_i, \boldsymbol{\epsilon}_i)$ according to Eq. (5)
    Update $\boldsymbol{\theta}, \boldsymbol{\eta}, \boldsymbol{\phi}$ with gradients, $t \leftarrow t + 1$
**until** convergence
**return:** $\boldsymbol{\theta}, \boldsymbol{\eta}, \boldsymbol{\phi}$

---

## B   Experiment details

### B.1   Graph neural networks

#### B.1.1   Model descriptions

As in Veličković et al. [7], we apply a two-layer GAT model. We summarize the graph attention layer here. Denote the input node features as $\boldsymbol{h} = \{\boldsymbol{h}_1, ..., \boldsymbol{h}_N\}$, where $N$ is the number of nodes. Then, the self-attention weights is defined as:

$$\alpha_{ij}^h = \frac{\exp(\mathrm{LeakyReLU}(\boldsymbol{a}^h[\mathbf{W}^h\boldsymbol{h}_i || \mathbf{W}^h\boldsymbol{h}_j]))}{\sum_{k \in \mathcal{N}_i} \exp(\mathrm{LeakyReLU}(\boldsymbol{a}^h[\mathbf{W}^h\boldsymbol{h}_i || \mathbf{W}^h\boldsymbol{h}_k]))},$$

where $\boldsymbol{a}^h, \mathbf{W}^h$ are neural network weights for head $h$, and $\mathcal{N}_i$ is the set of neighbor nodes for node $i$. $||$ denotes concatenation.

The output $\boldsymbol{h}' = \{\boldsymbol{h}'_1, ..., \boldsymbol{h}'_N\}$ is computed as:

$$\boldsymbol{h}'_i = ||_{h=1}^H \sigma\left(\sum_{j \in \mathcal{N}_i} \alpha_{ij}^h \mathbf{W}^h \boldsymbol{h}_j\right).$$

#### B.1.2   Detailed experimental settings

We follow the same architectural hyperparameters as in Veličković et al. [7]. The first layer consists of $H = 8$ attention heads computing 8 features each, and the second layer has a single head attention following an exponential linear unit (ELU) [67] nonlinearity. Then softmax is applied to obtain probabilities. During training, we apply $L2$ regularization with $\lambda = 0.0005$. Furthermore, dropout [41] with $p = 0.6$ is applied to both layers' inputs, as well as to the normalized attention coefficients. Pubmed requires slight changes for hyperparameter: the second layer has $H = 8$ attention heads, and the $L2$ regularization weight is $\lambda = 0.001$. Models are initialized using Glorot initialization [68] and trained with cross-entropy loss using the Adam SGD optimizer [69] with an initial learning rate of 0.01 for Pubmed, and 0.005 for all other datasets. In both cases we use an early stopping strategy on both the cross-entropy loss and accuracy on the validation nodes, with a patience of 100 epochs. Here, we summarize the hyperparameters for BAM, including anneal rate $\rho$ (as in Algorithm 1), $\sigma_1$ and $\sigma_2$ for prior Lognormal and posterior Lognormal respectively, $k$ for Weibull distribution, $\alpha, \beta$ for Gamma distribution, hidden dimension for contextual prior $d_{\mathrm{mid}}$. On Pubmed, we use anneal rate $\rho = 0.2$ for all methods. For BAM-LF, $\sigma_1 = 1\text{E}6$, $\sigma_2 = 1\text{E-}2$. For BAM-LC, $\sigma_1 = 1\text{E}5$, $\sigma_2 = 1\text{E-}2$, and $d_{\mathrm{mid}} = 5$. For BAM-WF, $k = 10$, $\beta = 1\text{E-}8$, $\alpha = 1\text{E-}4$. For BAM-WC, $k = 10$, $\beta = 1\text{E-}4$, and $d_{\mathrm{mid}} = 5$. On Cora, for BAM-LF, $\sigma_1 = 1\text{E}15$, $\sigma_2 = 1\text{E-}6$, and $\rho = 0.2$. For BAM-LC, $\sigma_1 = 1\text{E}15$, $\sigma_2 = 1\text{E-}15$, $\rho = 0.1$, and $d_{\mathrm{mid}} = 1$. For BAM-WF, $k = 1$, $\beta = 1\text{E-}10$, $\alpha = 1\text{E-}15$, and $\rho = 0.2$. For BAM-WC, $k = 1$, $\beta = 1\text{E-}10$, $\rho = 0.1$, and $d_{\mathrm{mid}} = 1$. On Citeseer, we use anneal rate 0.1 for all methods. for BAM-LF, $\sigma_1 = 1\text{E}15$ and $\sigma_2 = 1\text{E-}6$. For BAM-LC, $\sigma_1 = 1\text{E}15$, $\sigma_2 = 1\text{E-}5$, and $d_{\mathrm{mid}} = 1$. For BAM-WF, $k = 100$, $\beta = 1\text{E-}15$, and $\alpha = 1\text{E-}7$. For BAM-WC, $k = 100$, $\beta = 1\text{E-}15$, and $d_{\mathrm{mid}} = 1$.

Table 6: Basic statistics on datasets for node classification on graphs.

|  | CORA | CITESEER | PUBMED |
|---|---|---|---|
| #NODES | 2708 | 3327 | 19717 |
| #EDGES | 5429 | 4732 | 44338 |
| #FEATURES/NODE | 1433 | 3703 | 500 |
| #CLASSES | 7 | 6 | 3 |
| #TRAINING NODES | 140 | 120 | 60 |
| #VALIDATION NODES | 500 | 500 | 500 |
| #TEST NODES | 1000 | 1000 | 1000 |

## B.2 Visual question answering

### B.2.1 Uncertainty evaluation via PAvPU

We adopt hypothesis testing to quantify the uncertainty of a model's prediction. Consider $M$ posterior samples of predictive probabilities $\{\boldsymbol{p}_m\}_{m=1}^{M}$, where $\boldsymbol{p}_m$ is a vector with the same dimension as the number of classes. To quantify how confident our model is about its prediction, we evaluate whether the difference between the probabilities of the first and second highest classes (in terms of posterior means) is statistically significant with two-sample $t$-test.

With the output $p$-values and a given threshold, we can determine whether a model is certain about its prediction. Then, we evaluate the uncertain using the Patch Accuracy vs Patch Uncertainty metric [46] which is defined as $\text{PAvPU} = (n_{ac} + n_{iu})/(n_{ac} + n_{au} + n_{ic} + n_{iu})$, where $n_{ac}, n_{au}, n_{ic}, n_{iu}$ are the numbers of accurate and certain, accurate and uncertain, inaccurate and certain, inaccurate and uncertain samples, respectively. Since for VQA, each sample has multiple annotations, the accuracy for each answer can be a number between $0$ and $1$ and it is defined as $\text{Acc}(ans) = \min\{(\#\text{human that said } ans)/3, 1\}$. Then we generalize the PAvPU for VQA task accordingly:

$$n_{ac} = \sum_i \text{Acc}_i \text{Cer}_i, \ n_{iu} = \sum_i (1 - \text{Acc}_i)(1 - \text{Cer}_i),$$

$$n_{au} = \sum_i \text{Acc}_i (1 - \text{Cer}_i), \ n_{ic} = \sum_i (1 - \text{Acc}_i)(\text{Cer}_i),$$

where for the $i$th prediction $\text{Acc}_i$ is the accuracy and $\text{Cer}_i \in \{0, 1\}$ is the certainty indicator.

### B.2.2 Model descriptions

We use the state-of-the-art VQA model, MCAN [11], to conduct experiments. The basic component of MCAN is Modular Co-Attention (MCA) layer. The MCA layer is a modular composition of two basic attention units: the self-attention (SA) unit and the guided-attention (GA) unit, where the SA unit focuses on intra-modal interactions and GA unit focuses on inter-modal interactions. Both units follow the multi-head structure as in Vaswani et al. [2], including the residual and layer normalization components. The only difference is that in GA, the queries come from a different modality (images) than the keys and values (questions). By stacking MCA layers, MCAN enables deep interactions between the question and image features. We adopt the encoder-decoder structure in MCAN [11] with six co-attention layers.

### B.2.3 Detailed experimental settings

We conduct experiments on the VQA-v2 dataset, which is split into the training (80k images and 444k QA pairs), validation (40k images and 214k QA pairs), and testing (80k images and 448k QA pairs) sets. The evaluation is conducted on the validation set as the true labels for the test set are not publicly available [29], which we need for uncertainty evaluation. For the noisy dataset, we add Gaussian noise (mean $0$, variance $5$) to image features. We follow the hyperparameters and other settings from Yu et al. [11]: the dimensionality of input image features, input question features, and fused multi-modal features are set to be $2048$, $512$, and $1024$, respectively. The latent dimensionality in the multi-head attention is $512$, the number of heads is set to $8$, and the latent dimensionality for each head is $64$. The dropout rate is $0.1$. The size of the answer vocabulary is set to $N = 3129$ using the strategy in Teney et al. [70]. To train the MCAN model, we use the Adam optimizer [69] with $\beta_1 = 0.9$ and $\beta_2 = 0.98$. The learning rate is set to $\min(2.5t\text{E-5}, 1\text{E-4})$, where $t$ is the current epoch

number starting from 1. After 10 epochs, the learning rate is decayed by $1/5$ every 2 epochs. All the models are trained up to 13 epochs with the same batch size of 64. To tune the hyperparameters in BAM, we randomly hold out $20\%$ of the training set for validation. After tuning, we train on the whole training set and evaluate on the validation set. For BAM-LF, $\sigma_1 = 1E9$, $\sigma_2 = 1E\text{-}9$, and $\rho = 0.2$. For BAM-LC, $\sigma_1 = 1E9$, $\sigma_2 = 1E\text{-}9$, $\rho = 0.2$, and $d_{mid} = 20$. For BAM-WF, $k = 1000$, $\beta = 1E\text{-}2$, $\alpha = 1E\text{-}3$, and $\rho = 0.2$. For BAM-WC, $k = 1000$, $\beta = 1E\text{-}6$, $\rho = 0.1$, and $d_{mid} = 20$.

### B.2.4 More results

Question: What animal is next to the giraffe?
Annotation set: {'wildebeest', 'horse', 'cow', 'antelope', 'gazelle', 'tapir', 'antelope', 'mountain lion', 'antelope', 'horse'}
Soft answer: deer, p-value: 0.01
**BAM-WC answer: cow, p-value: 0.35**

Question: What number is on the batter's shirt?
Annotation set: {'25', '25', '25', '25', '25', '25', '25', '25', '25', '25'}
Soft answer: 15, p-value: 0.0
**BAM-WC answer: 25, p-value: 0.0**

Question: Is there mustard on the hot dog?
Annotation set: {'yes', 'yes', 'yes', 'yes', 'yes', 'yes', 'yes', 'yes', 'yes'}
Soft answer: yes, p-value: 0.48
**BAM-WC answer: yes, p-value: 0.0**

Figure 2: VQA visualization: we present three image-question pairs along with human annotations. We show the predictions and uncertainty estimates of different methods. We evaluate methods based on their answers and $p$-values and highlight the better answer in bold (most preferred to least preferred: correct certain > correct uncertain > incorrect uncertain > incorrect certain).

Table 7: Performance comparison of different attention modules on visual question answering.

| | Accuracy | | | | |
|---|---|---|---|---|---|
| Attention | Original Data | | | Noisy Data | |
| | ALL | Y/N / NUM / OTHER | | ALL | Y/N / NUM / OTHER |
| Soft | 66.95 | 84.55 / 48.92 / 58.33 | | 61.25 | 80.58 / 40.80 / 51.97 |
| BAM-LF | 66.89 | 84.46 / 49.11 / 58.24 | | 61.43 | 80.95 / 41.51 / 51.85 |
| BAM-LC | 66.93 | 84.58 / 49.05 / 58.24 | | 61.58 | 80.70 / 41.31 / 52.40 |
| BAM-WF | 66.93 | 84.55 / 48.84 / 58.32 | | 61.60 | 81.02 / 41.84 / 52.05 |
| BAM-WC | **67.02** | 84.66 / 48.88 / 58.42 | | **62.89** | 81.94 / 41.90 / 53.96 |

| | Uncertainty | | | | |
|---|---|---|---|---|---|
| Attention | Original Data | | | Noisy Data | |
| | ALL | Y/N / NUM / OTHER | | ALL | Y/N / NUM / OTHER |
| Soft | 70.04 | 83.02 / 56.81 / 63.66 | | 65.34 | 78.84 / 49.80 / 59.18 |
| BAM-LF | 69.92 | 82.79 / 56.87 / 63.58 | | 65.48 | 79.13 / 50.19 / 59.16 |
| BAM-LC | 70.14 | 83.02 / 57.40 / 63.71 | | 65.60 | 78.85 / 49.87 / 59.70 |
| BAM-WF | 70.09 | 83.00 / 56.85 / 63.78 | | 65.62 | 79.16 / 50.22 / 59.40 |
| BAM-WC | **71.21** | 83.95 / 58.12 / 63.82 | | **66.75** | 80.21 / 51.38 / 60.58 |

## B.3 Image captioning

### B.3.1 Model descriptions

We conduct experiments on an attention-based model for image captioning, Att2in, in Rennie et al. [10]. This model uses RNN as its decoder, and at each step of decoding, image features are aggregated using attention weights computed by aligning RNN states with the image features. Formally, suppose $I_1, ..., I_N$ are image features, $\boldsymbol{h}_{t-1}$ is the hidden state of RNN at step $t-1$. Then, the attention weights at step $t$ are computed by: $\boldsymbol{\alpha}_t = \text{softmax}(\boldsymbol{a}_t + b_\alpha)$, and $a_t^i = W \tanh(W_{aI} I_i + W_{ah} \boldsymbol{h}_{t-1} + b_a)$, where $W, W_{aI}, W_{ah}, b_\alpha, b_a$ are all neural network weights. Aggregated image feature $I_t = \sum_{i=1}^{N} \alpha_t^i I_i$

would then be injected into the computation of the next hidden state of RNN $h_t$ (see details in Rennie et al. [10]).

### B.3.2 Detailed experimental settings

We use the code from `https://github.com/ruotianluo/self-critical.pytorch` and conduct our experiments on the MS COCO dataset [44] that consists of 123,287 images. Each image has at least five captions. We use the standard data split from Karpathy and Fei-Fei [71], with 113,287 training, 5000 validation, and 5000 testing images. The vocabulary size $V$ is 9488 and the max caption length $T$ is 16. We replace the ResNet-encoded features in Rennie et al. [10] with bounding box features extracted from a pre-trained Faster-RCNN [48] as visual features. Following the original setting in the code base, we use batch size 10, Adam optimizer with learning rate 5E-4, dropout rate of 0.5 and train 30 epochs. During training, we use MLE loss only without scheduled sampling or RL loss. For testing, we use greedy search to generate sequences. For BAM, we use contextual prior with $d_{\mathrm{mid}} = 10$ and $\rho = 1$. For BAM-WC, $k = 10$, $\beta =$ 1E-6. For BAM-LC, $\sigma_1 =$ 1E3, $\sigma_2 = 0.1$.

### B.4 Neural Machine Translation

### B.4.1 Model descriptions

To make comparision with Deng et al. [29], we adopt the LSTM-based machine translation model in that paper. The model uses a bidirectional LSTM to encode a source sentence to source representations $x_1, ..., x_T$. At the step $j$ of decoding, current LSTM state $\tilde{x}$ (a function of previous target words $y_{1:j-1}$) is used as query. The attention weights is computed from an MLP between the query and encoded source token representations. Then the aggregated feature is used to produce the distribution over the next target work $y_j$ (see details in Deng et al. [29] or see code in `https://github.com/harvardnlp/var-attn`).

### B.4.2 Detailed experimental settings

We use the same dataset, IWSLT [72], as Deng et al. [29]. We preprocess the data in the same way: using Byte Pair Encoding over the combined source/target training set to obtain a vocabulary size of 14,000 tokens [73]. We train on the sequence length up to 125. We use a two-layer bi-dreictional LSTM with 512 units and 768 units for the encoder and decoder, respectively. In addition, the batch size is 6, dropout rate is 0.3, learning rate is 3E-4 (Adam optimizer). For testing, we use beam search with beam size 10 and length penalty 1 [74]. For BAM-WC, $k = 5$, $\beta =$ 1E-6, $\rho = 1$, and $d_{\mathrm{mid}} = 5$.

Table 8: Step time (sec) and number of parameters of variational attention [26] and BAM on NMT.

|  | s/step | params |
|---|---|---|
| VA-Enum | 0.12 | 64M |
| VA-Sample | 0.15 | 64M |
| BAM-WC | **0.10** | **42M** |

### B.5 Pretrained language models

### B.5.1 Model descriptions

BERT [3] is a state-of-the-art deep bidirectional transformer[2] model pretrained on large corpora to extract contextual word representations. ALBERT [4] improves upon BERT in terms of latency efficiency and performance by using (a) factorized embedding parameterization, (b) cross-layer parameter sharing, and (c) a sentence-order prediction (SOP) loss. Our experiment is done on the ALBERT-base model, which includes 12 attention layers, each of hidden dimension 768. The embedding dimension for factorized embedding is 128. While BERT-base involves $108M$ parameters, ALBERT-base only has $12M$ parameters.

### B.5.2 Detailed experimental settings

Our experiment includes both the General Language Understanding Evaluation (GLUE) and Stanford Question Answering (SQuAD) Datasets. We evaluate on 8 tasks from GLUE including Corpus

of Linguistic Acceptability (CoLA; [75]), Stanford Sentiment Treebank (SST; [76]), Microsoft Research Paraphrase Corpus (MRPC; [77]), Semantic Textual Similarity Benchmark (STS;[78]), Quora Question Pairs (QQP; [79]), Multi-Genre NLI (MNLI; [80]), Question NLI (QNLI; [60]), and Recognizing Textual Entailment (RTE; [81]). We evaluate on both SQuAD v1.1 and SQuAD v2.0. Our code is built on Wolf et al. [62], which can be found at `https://github.com/huggingface/transformers`. We follow the training settings as in Lan et al. [4] and summarize them in Table 9. We also include the hyperparameter setting for BAM-WC. We note, as the model is already pretrained so we do not anneal KL term. We pick $\beta = 1\text{E-}2$ and $d_{\text{dim}} = 5$ for all experiments, as we found the performance is not sensitive to them. We include the $k$ in Table 9.

Table 9: Experiment setting for pretrained language model (LR: learning rate, BSZ: batch size, DR: dropout rate, TS: training steps, WS: warmping steps, MSL: maximum sentence length).

|  | LR | BSZ | ALBERT DR | CLASSIFIER DR | TS | WS | MSL | $k$ |
|---|---|---|---|---|---|---|---|---|
| CoLA | 1.00 E-05 | 16 | 0 | 0.1 | 5336 | 320 | 512 | 10 |
| STS | 2.00 E-05 | 16 | 0 | 0.1 | 3598 | 214 | 512 | 20 |
| SST-2 | 1.00 E-05 | 32 | 0 | 0.1 | 20935 | 1256 | 512 | 1000 |
| MNLI | 3.00 E-05 | 128 | 0 | 0.1 | 10000 | 1000 | 512 | 5 |
| QNLI | 1.00 E-05 | 32 | 0 | 0.1 | 33112 | 1986 | 512 | 500 |
| QQP | 5.00 E-05 | 128 | 0.1 | 0.1 | 14000 | 1000 | 512 | 1000 |
| RTE | 3.00 E-05 | 32 | 0.1 | 0.1 | 800 | 200 | 512 | 1000 |
| MRPC | 2.00 E-05 | 32 | 0 | 0.1 | 800 | 200 | 512 | 100 |
| SQuAD v1.1 | 5.00 E-05 | 48 | 0 | 0.1 | 3649 | 365 | 384 | 10 |
| SQuAD v 2.0 | 3.00 E-05 | 48 | 0 | 0.1 | 8144 | 814 | 512 | 2000 |