[Reviews · NeurIPS 2020]

Review 1

Summary and Contributions: This paper proposes to treat attentions as continuous latent variables, where training uses VAE. To parameterize the prior and approxiamte posterior, this work uses reparametrizable distributions such as Weibull and lognormal distributions to get unnormalized weights, and then normalizes them to get attentions. This procedure has lower variance due to using pathwise gradients. In order to compute the KL term, this work further approximates KL with the KL between the untransformed distributions which have analytic forms. Or equivalently, such an approximation can also be viewed as modeling the unnormalized attentions as latent variables. Parameterization of the approximate posterior is also different from prior works: the approximate posterior directly comes from the soft attention weights, which makes the parameterization light-weight (instead of relying on a separate inference network), but at the same time, the inference network does not get access to the entire target, so it can never reach the true posterior. Experiments show that the proposed continuous latent attention mechanism gets better performance compared to deterministic attention on a wide variety of tasks, including image captioning, machine translation, graph classification, and finetuning BERT.

Strengths: 1. This work enables modeling uncertainties of the attention variable, complementing prior works on discrete latent attentions. 2. The proposed method is simple, does not require s separate inference network and can be easily plugged to existing models using attentions. 3. Empirically the results are better than determinstic attentions on a variety of tasks.

Weaknesses: 1. The improvement compared to deterministic attention seems marginal on some tasks such as VQA and machine translation. 2. It is not very clear what benefits modeling attention uncertainties give us. Unlike prior works on discrete latent variables that can make interpretability claims, I'm not sure why we want to model soft attention as a latent variable. Are they better under low resource scenarios? Also, can you do more quantification of the benefits of modeling attention uncertainties? Or even qualitatively showing a few examples of samples from the attention distribution and see if they truly reflect the underlying uncertainties. 3. Related to 2, what's the benefit of latent continuous attention over latent discrete attention? The hard attention baseline looks too weak to me, probably a Gumbel-Softmax type discrete attention is a more reasonable baseline. 4. The inference network does not get the true outputs, so the "approximate posterior" can never reach true posteriors. I'm not sure why you need a separate prior in this formulation. IMHO the "approximate posterior" here shall be used as the prior (it is how it's used at test time), which immediately sets the KL term to zero hence the ELBO is better than optimizing with a different prior. This would be the continuous version of Xu et al 2015's "hard attention". Why would using a different prior help (to me it gets the same information as the "approximate posterior" here)? Can you do an ablation study where you get rid of the KL term in Eq. 4? 5. While I can understand that the true KL between attention distributions is not computable, the approximation here is essentially shifting the latent variable modeling from attentions to the unnormalized weights. However, using the "approximate posterior" as the prior gets rid of the KL hence this is no longer a problem. ===post-rebuttal=== My concern about setting prior to the approximate posterior has been addressed in the authors' response. It might be useful to incorporate it into the next version.

Correctness: Yes.

Clarity: Yes, it's very easy to follow.

Relation to Prior Work: Yes, to the best of my knowledge.

Reproducibility: Yes

Additional Feedback: I think using a separate prior requires more justification. An easier and more parameter efficient way is to simply use the "approximate posterior" here as the prior, resulting a continuous version of "hard attention".


Review 2

Summary and Contributions: This paper proposes a stochastic version of attention (BAM) with differentiable ELBO object. The experiments show competitive results against strong baseline methods. The variational distributions can be Weibull or Lognormal which are reparameterizable. It makes the stochastic soft attention practical.

Strengths: 1. Introduce a context related soft attention model by introducing a data dependent prior $p_{\eta}(W)$, where a prior beliefs of attention distribution can be injected. 2. BAM use reparameterizable variational distribution as an approximation to the posterior of attention weights. It makes the training process under the framework of gradient descent. 3. BAM is applied to various attention based models (graph node classification, QVA, NLP, etc.), and the experiments show its improvement.

Weaknesses: The variational distribution is a certain distribution rather than a family. This potentially restricted the expression ability of the attention. Compared with existing method, the improvement comes from different variational distribution and reparameterization trick.

Correctness: yes

Clarity: yes

Relation to Prior Work: yes

Reproducibility: Yes

Additional Feedback: As mentioned in the paper, the BAM is more efficient than existing method, could you please provide the running time of BAM and comparison methods. This will give me a quantitative perception.


Review 3

Summary and Contributions: The paper proposes a Bayesian soft-attention module of general use and efficient in terms of memory and computational cost

Strengths: 1. Stochastic attention modules that allow to perform Bayesian inference for neural network learning, which in turn allow to produce posterior distribution predictions rather than point estimates 2. Extensive study over different domains and problems 3. Efficient design that can be easily extended from the existing attention models

Weaknesses: Good paper, the following are some notes to further improve the paper rather than major weaknesses: 1. Some discussion on why posterior inference is chosen over W among all the parameters existing in attention network would be beneficial 2. It would be interesting to know why the gamma distribution itself was not chosen instead of the Weibull distribution 3. In experiments it would be interesting to see more comparison with other stochastic attention methods. For example, Deng, Y., Kim, Y., Chiu, J., Guo, D. and Rush, A., 2018. Latent alignment and variational attention. In Advances in Neural Information Processing Systems (pp. 9712-9724) (mentioned in the related work) also used the same dataset for the VQA task, but it is only considered for the machine translation task 4. It is unclear why uncertainty estimation is evaluated only on one task 5. Table 1 ā€“ why does only BAM-WC have standard deviation results?

Correctness: The claims and empirical methodology appear to be correct

Clarity: The paper is very well written and easy to follow. I enjoy reading it. A couple of comments: 1. Figure 1 requires a bit more explanation 2. Line 299, MLE is not defined 3. Line 301, missing reference for evaluation metrics, couple of words on 4 different BLEU metrics in Table 3 would be appreciated Minor: Eq. (1) ā€“ in the denominator it is better to use a different notation for j (e.g. jā€™) to avoid confusion with j on the left-hand side and in the numerator

Relation to Prior Work: It is clearly discussed how this work differs from previous contributions

Reproducibility: Yes

Additional Feedback: Update after the response: I have read the other reviews and authors' response. I would like to thank the authors for their replies. I remain with the opinion that this is a good paper worthy to be presented at NeurIPS =========================================== See weaknesses

[Author Response · NeurIPS 2020]

We thank all reviewers for their valuable feedback. Below please find our response to each individual review.

**(R1)** *Significance of improvements*: For VQA, with provided error bars, the improvements are statistically significant.
Especially in the more challenging noisy scenario, the improvements
are over 1 point which is over 10 times the standard deviation. For
NMT, the error bar for BAM-WC is 0.02 (we will add it to the revi-
sion), so the improvement is also statistically significant. Meanwhile,
in Table S1, we compare the run time and number of parameters
of BAM and variational attention [26], where we show that BAM
achieves better results while being more efficient in time and memory.

Table S1: Step time (sec) and number of parameters of variational attention [26] and BAM on NMT.

|            | S/STEP | PARAMS |
|------------|--------|--------|
| VA-ENUM    | 0.12   | 64M    |
| VA-SAMPLE  | 0.15   | 64M    |
| BAM-WC     | **0.10** | **42M** |

*Benefits of modeling attention weights as continuous latent variables*: (a) Modeling attention weights as latent
variables enhances the model's ability to cap-
ture complicated dependencies and calibrate
uncertainty, and prevents overfitting due to
the added randomness. To evaluate the uncer-
tainty quantitatively, we provide the PAvPU
results for VQA in Table 1 (main paper) and
for graph in Table S3. To evaluate it quali-
tatively, we visualize the predictions and un-
certainties of three VQA examples in Fig-
ure S1. (b) Compared to previous work us-
ing discrete latent variables, using continuous
ones is much easier to optimize. Also, BAM
is faster and demands less memory and hence
more suitable for low resource scenarios. On
hard attention baseline, the choice of REIN-
FORCE gradient estimator is based on previ-

*Question: What animal is next to the giraffe? Annotation set: {'wildebeest', 'horse', 'cow', 'antelope', 'gazelle', 'tapir', 'antelope', 'mountain lion', 'antelope', 'horse'} Soft answer: deer, p-value: 0.01* **BAM-WC answer: cow, p-value: 0.35**

*Question: What number is on the batter's shirt? Annotation set: {'25', '25', '25', '25', '25', '25','25', '25, '25'} Soft answer: 15, p-value: 0.0* **BAM-WC answer: 25, p-value: 0.0**

*Question: Is there mustard on the hot dog? Annotation set: {'yes', 'yes', 'yes', 'yes', 'yes', 'yes', 'yes', 'yes', 'yes', 'yes'} Soft answer: yes, p-value: 0.48* **BAM-WC answer: yes, p-value: 0.0**

Figure S1: VQA visualization: we present three image-question pairs along with human annotations. We show the predictions and uncertainty estimates of different methods. We evaluate methods based on their answers and $p$-values and highlight the better answer in bold (most preferred to least preferred: correct certain > correct uncertain > incorrect uncertain > incorrect certain).

ous work [9, 26]. In [26], Gumbel-softmax, which provides biased gradients, was found to underperform REINFORCE
gradient estimator for NMT, which is why we have not included it in the experiment.

*Ablation study on prior distributions*: It is true that the inference network does not get the entire targets during
training, which is the reason that it can be used at the test time to help predict
the outputs. We introduce a contextual prior distribution to impose further
regularization on the attention distributions. We agree if setting the prior and
variational posterior the same, the KL in ELBO vanishes and regularization
disappears. We have added an ablation study accordingly, as shown in Table S2,
which suggests the importance of appropriate KL regularization. We will add them into the paper in revision.

Table S2: Accuracy for graphs.

| ATTENTION | CORA | CITESEER | PUBMED |
|-----------|------|----------|--------|
| GAT | 83.00 | 72.50 | 77.26 |
| BAM (REMOVE KL) | 83.39 | 72.91 | 78.50 |
| BAM-WC | **83.81** | **73.52** | **78.82** |

**(R2)** (1) We clarify that the proposed BAM framework works for any reparameterization distribution defined on the
non-negative real line, and we have chosen Weibull and Lognormal from this family as representative examples. (2)
Compared with existing method, BAM is different in not only variational distribution and gradient estimation, but also
prior distribution. (3) Table S1 shows that BAM is more efficient in both time and memory than variation attention [26].

**(R3)** (1) Modeling attention weights is a quite standard approach [9,26,28] as they have intuitive meanings.
Also in BAM, the attention weights are data dependent local variables. This approach is more computa-
tionally efficient compared to the convention in Bayesian neural network where neural network parameters,
such as $\theta$, are modeled as globally shared random variables (i.e., not data dependent), as the latter approach
needs multiple sampled sets of NN weights to provide uncertainty estimation.
In BAM, we only need a single set of global parameters (NN weights) and rely
on the stochasticity on $W$ to provide uncertainty. (2) We did not choose the
gamma distribution as it is not reparameterizable and hence pathwise gradients
that are unbiased and have low variance are not available. (3) We did not include

Table S3: PAvPU for graphs.

| ATTENTION | CORA | CITESEER | PUBMED |
|-----------|------|----------|--------|
| GAT | 82.30 | 72.80 | 77.20 |
| BAM-WC | **83.50** | **73.90** | **78.10** |

comparison with [26] in VQA as we used a better baseline model than theirs that had already provided better performance.
As their method requires a case by case design, it is unclear to us how to adapt their method to our model. Further, the
code of [26] was only available for NMT so we only include the comparison for NMT. (4) In Table S3, we have included
the uncertainty estimation result in terms of PAvPU for graph node classification as well and observed consistent
improvements. For other tasks like image captioning, it is unclear to us how to evaluate uncertainty. (5) In the paper, we
eliminated some error bar trying to prevent the table from being too crowded. We will add them in revision. *Other*
*comments:* In our revision, we will add more detailed explanation for Figure 1, include the definition of MLE and
BLEU metrics, incorporate the missing reference for evaluation metrics, and update the notation of Equation 1.

[Meta-Review · NeurIPS 2020]

This paper proposes considering attention mechanisms as continuous latent variables, using VAEs for training. It uses reparametrizable distributions such as Weibull and log-normal distributions to get unnormalized weights, which are then normalized. Experiments show that the proposed continuous latent attention mechanism gets better performance compared to deterministic attention on a wide variety of tasks, including image captioning, machine translation, graph classification, and fine-tuning BERT. All reviewers recommended acceptance, pointing out that this is an interesting idea and a solid and well-executed work. One concern was raised about the significance of improvement on VQA and NMT, and about directly setting prior to approximate posterior, which the authors addressed in the rebuttal. I agree with the reviewers and recommend acceptance. However, I encourage the authors to follow the reviewers’ suggestions to improve the paper, including discussing the advantages/disadvantages of continuous latent variable models over discrete latent variable models, as suggested by R1.